# DETECTING OUT-OF-DISTRIBUTION SAMPLES VIA CONDITIONAL DISTRIBUTION ENTROPY WITH OPTIMAL TRANSPORT

## ABSTRACT

When deploying a trained machine learning model in the real world, it is inevitable to receive inputs from out-of-distribution (OOD) sources. For instance, in continual learning settings, it is common to encounter OOD samples due to the non-stationarity of a domain. More generally, when we have access to a set of test inputs, the existing rich line of OOD detection solutions, especially the recent promise of distance-based methods, falls short in effectively utilizing the distribution information from training samples and test inputs. In this paper, we argue that empirical probability distributions that incorporate geometric information from both training samples and test inputs can be highly beneficial for OOD detection in the presence of test inputs available. To address this, we propose to model OOD detection as a discrete optimal transport problem. Within the framework of optimal transport, we propose a novel score function known as the *conditional distribution entropy* to quantify the uncertainty of a test input being an OOD sample. Our proposal inherits the merits of certain distance-based methods while eliminating the reliance on distribution assumptions, a-prior knowledge, and specific training mechanisms. Extensive experiments conducted on benchmark datasets demonstrate that our method outperforms its competitors in OOD detection.

## 1 INTRODUCTION

Training a machine learning model often assumes that the training samples and test inputs are drawn from the same distribution. However, when deploying a trained model in the open-world, out-of-distribution (OOD) inputs, which come from a different distribution of training samples, i.e., in-distribution (ID) samples, are inevitable. For example, in continual learning settings, with the inherent non-stationarity of a domain, it's typical to observe samples in a test setting which are Out-Of-Distribution (OOD) w.r.t. the training set (Garg et al., 2023). The ignorance or overconfidence with OOD inputs leads to unwanted model predictions, Therefore, a trustworthy machine learning model should keep a sharp lookout for OOD and ID inputs.

A flurry of works (Hendrycks & Gimpel, 2017; Sastry & Oore, 2020; Fang et al., 2022; Fort et al., 2021; Liu et al., 2020; Nandy et al., 2020) has been proposed on OOD detection; recent research attention is drawn by distance-based OOD detection (Sehwag et al., 2021; Sun et al., 2022) for its promising performance. Distance-based methods utilize some metrics (Weinberger & Saul, 2009; Kulis et al., 2013) in the feature representation space for differentiating OOD inputs from ID samples, with the built-in assumption that an OOD sample stays relatively far from ID samples. For example, some works (Sehwag et al., 2021; Ming et al., 2023) use statistical information (e.g., mean, variance) of in-distribution and calculate the Mahalanobis distance as the score function. The performance depends on the fitness between the parameterized Gaussian distribution and real distribution of the training data (Morteza & Li, 2022; Maciejewski et al., 2022). However, the distributional assumption may not hold in some non-stationarity scenarios, such as continual learning or domain adaption, presenting a pitfall in outlier detecting and thus giving rise to the risk of inconsistent estimation. To alleviate it, (Sun et al., 2022) propose to only use $k$-th nearest training sample of a test input as the score function for OOD detection. With simplicity and efficacy, the method roots in pair-wise distance comparison, leaving untapped potential for exploiting population-wise information, as exposed in

continual learning settings, where a set of accessible test inputs is provided. The above limitations motivate us to study the following question:

*Can we leverage the empirical distributions of both training samples and test inputs with geometric information to discriminate out-of-distribution data?*

In this paper, we claim that the empirical distributions incorporating geometric information are beneficial for OOD detection in the presence of a set of accessible test inputs. This idea presents a significant departure from previous works in several key aspects. First, the idea focus on analyzing distributional discrepancies while incorporating geometric structure information in the feature space, which distinguishes it from the pair-wise distance comparison and allows it to leverage population-wise information for improving the performance. Second, the empirical distribution refers to the observed data distribution rather than the one conforming to a hypothesis, potentially eliminating the distribution assumption (e.g., multivariate Gaussian distribution assumption in Mahalanobis distance). Third, through the consideration of multiple test inputs, the mutual benefits of test inputs in OOD detection can be fully exploited, which has good potentials in some scenarios, such as continual learning and domain adaption, where we have access to a set of test inputs or even the entire test data.

Following this line of thoughts, we hereby propose a novel OOD detection method based on optimal transport theory. We construct empirical probability measures for training samples and test inputs, which lifts the Euclidean space of feature representations to a probability space. There are two advantages of doing so: 1) the empirical probability measures utilize the distribution information without assumptions about the underlying distribution; 2) it enables the measurement of the discrepancy between probability measures, which captures the significant difference between their corresponding supports, representing (training and test) samples. Combining pair- and population-wise information, optimal transport provides a geometric way to measure the discrepancy between empirical probability measures, making a basis for discriminating OOD samples.

Then, to measure the uncertainty of a test input being an OOD sample, we propose a novel score function, called as *conditional distribution entropy*. The sensitivity of conditional distribution entropy in capturing OOD inputs is enabled by the paradigm of mass split under marginal constraints in discrete optimal transport, where the mass of the OOD input transported to training samples is dominated by that of ID test inputs. In particular, an ID input with a certain conditional transport plan corresponds to a low conditional distribution entropy, while an OOD input with an uncertain conditional transport plan corresponds to a high conditional distribution entropy.

We conduct extensive experiments on benchmark datasets to gain the insights into our proposals. The results show that our proposed method is superior to the baseline methods. In particular, on challenging tasks, such as CIFAR-100 vs. CIFAR-10, our method achieves $14.12\%$ higher AUROC and $11.12\%$ higher AUPR than the state-of-the-art solution KNN+.

## 2 PRELIMINARIES

### 2.1 PROBLEM DEFINITION

**Definition 2.1** (OOD Detection). Let $\mathbb{X}$ be the sample space and $\mathbb{Y} = \{1, 2, ..., K\}$ be the label space. Given a training dataset $\mathcal{D}_{tr}^{in}$, we assume data is sampled from the joint distribution $\mathcal{P}_{\mathbb{XY}}^{in}$ over the joint space $\mathbb{X} \times \mathbb{Y}$. A trustworthy machine learning model is expected to not only accurately predict on known ID inputs, but also identify unknown OOD inputs (Sun et al., 2022). In the open world, test inputs $\mathcal{D}_{te}$ consists of both ID and OOD data. Given a set of test inputs from $\mathcal{P}_{\mathbb{X}}^{in} \times \mathcal{P}_{\mathbb{X}}^{ood}$, the goal of OOD detection is to identify whether an input $\mathbf{x} \in \mathbb{X}$ is from ID ($\mathcal{P}_{\mathbb{X}}^{id}$) or OOD ($\mathcal{P}_{\mathbb{X}}^{ood}$).

### 2.2 OPTIMAL TRANSPORT

The optimal transport (Villani, 2003) is to seek an optimal transport plan between two probability measures at the minimal cost, measured by a Wasserstein distance. The basic format is shown as follows. More details can be found in (Peyré & Cuturi, 2018).

**Wasserstein Distance.** Let $\mathcal{S}$ be a locally complete and separable metric space, $\mathcal{P}(\mathcal{S})$ be a Borel probability measure set on $\mathcal{S}$. For any $\mathcal{X}, \mathcal{X}' \subset \mathcal{S}$, assuming probability measures $\mu \in \mathcal{P}(\mathcal{X})$ and

$\nu \in \mathcal{P}(\mathcal{X}')$, the optimal transport defines a Wasserstein distance between $\mu$ and $\nu$, formulated as

$$W_p(\mu, \nu) := \left( \inf_{\pi(\mu,\nu)} \int_{\mathcal{X} \times \mathcal{X}'} ||x - x'||^p d\pi(\mu, \nu) \right)^{\frac{1}{p}} \tag{1}$$

$p \geq 1$, where the $\pi(\mu, \nu)$ is the set of joint probability measures with marginals $\mu$ and $\nu$.

**Discrete Optimal Transport.** Let $\Delta_n = \{\boldsymbol{\alpha} \in \mathbb{R}_+^n | \sum_{i=1}^n \alpha_i = 1, \forall \alpha_i \geq 0\}$ be an $n$-dimensional probability simplex. Consider two empirical probability measures $\mu = \sum_{i=1}^n \alpha_i \delta_{x_i}$ and $\nu = \sum_{j=1}^m \beta_j \delta_{x'_j}$, defined on metric spaces $\mathcal{X}$ with support $\{x_i\}_{i=1}^n$ and $\mathcal{X}'$ with support $\{x'_j\}_{j=1}^m$, respectively. Here, the weight vector $\boldsymbol{\alpha} = (\alpha_1, \alpha_2, ..., \alpha_n)$ and $\boldsymbol{\beta} = (\beta_1, \beta_2, ..., \beta_m)$ live in $\Delta_n$ and $\Delta_m$, respectively. The $\delta$ stands for the Dirac unit mass function. Then, given a transport cost $c : \mathcal{X} \times \mathcal{X}' \to \mathbb{R}_+$, the discrete optimal transport between probability measures $\mu$ and $\nu$ can be formalized as:

$$W_p^p(\mu, \nu) := \min_{\mathbf{P} \in \Pi(\mu,\nu)} \langle \mathbf{C}, \mathbf{P} \rangle_F \quad s.t. \ \mathbf{P}\mathbf{1}_m = \mu, \ \mathbf{P}^T \mathbf{1}_n = \nu \tag{2}$$

where $\mathbf{C} \in \mathbb{R}_+^{n \times m}$ is the transport cost matrix, and element $\mathbf{c}_{ij}$ represents a unit transport cost from $x_i$ to $x'_j$. The $\mathbf{P} \in \mathbb{R}_+^{n \times m}$ is the transport plan and $\mathbf{P}^T$ is the transpose of $\mathbf{P}$. The $\mathbf{1}$ denotes the all-ones vector. All feasible transport plans constitute the transport polytope $\Pi(\mu, \nu)$. The $\langle \mathbf{C}, \mathbf{P} \rangle_F$ is the Frobenius inner product of matrices, which equals to $tr(\mathbf{C}^T \mathbf{P})$.

## 3 METHOD

**Overview.** In this section, we study the method for OOD detection based on optimal transport theory. In Section 3.1, we first construct empirical probability measures for training samples and test inputs; then transform the problem of distributional discrepancy between probability measures as the problem of the entropic regularized optimal transport, yielding the optimal transport plan. In Section 3.2, we study how to use the optimal transport plan for modeling the score function for OOD detection. In Section 3.3, we investigate how to leverage supervised contrastive training (Khosla et al., 2020) to extract compact feature representations. Lastly, we extend our proposal to the unsupervised setting, in presence of training data without labels, which shows the generality of our proposal. The framework of our method is shown in Figure 4 (see Appendix A).

### 3.1 OPTIMAL TRANSPORT FOR OOD DETECTION

**Feature Extraction.** Given a training dataset with $N$ samples $\mathcal{D}_{tr}^{in} = \{(\mathbf{x}_i, y_i)\}_{i=1}^N$, where $\mathbf{x}_i$ represents the $i$-th sample and $y_i$ denotes the corresponding label. The functionality of the feature extraction can be represented as a function $f \colon \mathbb{X} \to \mathbb{V}$ that maps an input sample from the $n$-dimensional input space $\mathbb{X} \subseteq \mathbb{R}^n$ to a $d$-dimensional feature space $\mathbb{V} \subseteq \mathbb{R}^d$. In this way, we obtain a set of feature representations $\{f(\mathbf{x}_i)\}_{i=1}^N$ of the training samples, which are used for the subsequent tasks of OOD detection.[1]

**Empirical Probability Measure.** After the features are extracted, to utilize empirical distributions, we first construct a probability measure $\mathcal{P}$ over the low-dimensional feature space $\mathbb{V}$, which potentially lifts the Euclidean feature space $\mathbb{V}$ to a probability space $\mathbb{P} = \{\mathbb{V}, \sigma, \mathcal{P}\}$, where $\sigma$ denotes a well-defined algebra structure (Tao, 2011). Let the cardinality of the training dataset $\mathcal{D}_{tr}^{in}$ and test inputs $\mathcal{D}_{te}$ be $N$ and $M$, respectively, we define the discrete empirical probability measures of $\mathcal{D}_{tr}^{in}$ and $\mathcal{D}_{te}$ as $\mu$ and $\nu$, respectively, which are formulated as:

$$\mu = \sum_{i=1}^N \alpha_i \delta_{x_i} \quad \nu = \sum_{j=1}^M \beta_j \delta_{x'_j},$$

where $\delta$ is a Dirac unit mass function and $x_i$ (likewise for $x'_j$) denotes $i$-th feature representation $f(\mathbf{x}_i)$, i.e., the support of probability measure $\mu$. For simplicity, we use uniform weights for $\boldsymbol{\alpha}$ and $\boldsymbol{\beta}$.

---

[1] We normalize the features to a hypersphere, where inner production or cosine distance between feature vectors are natural choices. Please refer to Section 3.3 for more details.

---

**Algorithm 1** Conditional Distribution Entropy Score Function

---

**Input:** probability measure $\mu$ and $\nu$, cost matrix $\mathbf{C}$, regularization coefficient $\lambda$
Initialize $\mathbf{K} = exp(-\mathbf{C}/\lambda)$, $\mathbf{u} \in \mathbb{R}^N_+$, $\mathbf{v} \in \mathbb{R}^M_+$
**while** $(\mathbf{u},\mathbf{v})$ not converged **do**
    $\mathbf{u} = \mu \oslash (\mathbf{Kv})$ $\{\oslash$: point-wise division$\}$
    $\mathbf{v} = \nu \oslash (\mathbf{K}^T\mathbf{u})$
**end while**
$\mathbf{P}^\star \leftarrow diag(\mathbf{u})\mathbf{K}diag(\mathbf{v})$
Initialize Res with a null array
**for** $i = 1$ to $M$ **do**
    Res $\leftarrow$ condEntropy($col_i(\mathbf{P})$) $\{$append$\}$
**end for**

---

**Entropic Regularized Optimal Transport.** The constructed probability measures on the training samples and test inputs enables the measurement of the distributional discrepancy. Then, the problem is how distance information can be encoded in association with the discrepancy of probability measures to get both merits. As mentioned in Section 2.2, optimal transport provides a geometric manner to compare probability measures, paving the road for measuring both pair- and population-wise information. However, conventional optimal transport incurs cubic time complexity, which is prohibitive in real applications. To tackle the challenge, we formulate the OOD detection problem as a discrete optimal transport problem with entropic regularization, denoted as,

$$\mathcal{L}_\lambda(\mu, \nu, \mathbf{C}) = \min_{\mathbf{P}\in\Pi(\mu,\nu)} \langle \mathbf{C}, \mathbf{P} \rangle_F - \lambda E(\mathbf{P}) \; s.t. \; \mathbf{P1}_M = \mu, \; \mathbf{P}^T\mathbf{1}_N = \nu, \; \lambda \geq 0 \qquad (3)$$

where $E = -\sum_{ij} p_{ij}log(p_{ij} - 1)$ is the entropy function of transport plan formalized in Definition. 3.1, and $\mathbf{C} \in \mathbb{R}^{N\times M}$ is the matrix of pairwise cosine distances[1] between the supports. The element $p_{ij}$ of transport plan $\mathbf{P} \in \mathbb{R}^{N\times M}$ denotes the mass transported from probability measure $\mu_i$ to $\nu_j$. By solving the problem above, we can obtain an optimal transport plan $\mathbf{P}^\star$, as described in Theorem 3.2.

**Definition 3.1.** *Suppose two discrete random variables $U \sim \mu$ and $V \sim \nu$, following $(U, V) \sim \pi(\mu,\nu)$, where $\pi(\mu,\nu)$ is the joint distribution with marginals $\mu$ and $\nu$. The joint entropy of random variables $U$ and $V$ is defined as:*

$$\mathbb{H}(U, V) = -\sum_i \sum_j \pi_{ij}log(\pi_{ij})$$

The above definition indicates that transport plan $P$ is essentially a joint distribution with marginals $\mu$ and $\nu$.

**Theorem 3.2.** *The problem $\mathcal{L}_\lambda(\mu, \nu, \mathbf{C})$ has a unique optimal solution.*

*Proof sketch.* The entropy function of transport plan $E(\mathbf{P})$ is a concave function, which can be evidenced by computing the Hessian matrix with regard to the transport plan. Thus, the problem is $\lambda$-strongly convex w.r.t. the transport plan, and therefore has a unique optimal solution.

Theorem 3.2 shows that solving the discrete entropic regularization optimal transport problem Equation 3, we can obtain a unique optimal transport plan $\mathbf{P}^\star$. For more details, we refer to (Boyd et al., 2004; Bertsimas & Tsitsiklis, 1997).

**Entropic Regularized OT in the Dual.** The reason we study the dual problem of entropic regularized optimal transport is the dual problem allows us to obtain an optimal transport plan via matrix scaling (Sinkhorn & Knopp, 1967) for the primal problem (Equation 3), which is computationally efficient (Cuturi, 2013). Moreover, the optimal transport plan can be used for modeling the conditional distribution entropy score function discussed in Section 3.2.

**Proposition 3.3.** *The optimal transport plan of the dual problem has the form:*

$$\mathbf{P}^\star_{ij} = e^{(u_i-\mathbf{C}_{ij}+v_j)/\lambda} = \mathbf{u}_i\mathbf{K}_{ij}\mathbf{v}_j,$$

*where the introduced $\mathbf{u}$ and $\mathbf{v}$ are dual variables, and $\mathbf{K} = e^{-\mathbf{C}/\lambda}$.*

*Proof sketch.* With introducing Lagrangian associated to the primal problem (Equation 3), we can transform the primal problem with marginal constraints into an unconstrained optimization problem regarding the transport plan and dual variables. Then, taking derivatives on the transport plan leads to the above result.

Propostion 3.3 indicates that the optimal transport plan has the matrix multiplication form and satisfies marginal constraints. Thus, it can be regarded as a matrix scaling problem and can be solved with Sinkhorn iteration (Sinkhorn & Knopp, 1967) in quadratic time complexity.

### 3.2 CONDITIONAL DISTRIBUTION ENTROPY MEASURES OOD SAMPLES

In this section, we introduce a novel score function derived from the optimal transport plan to examine whether an input sample is OOD or not. The transport plan is a joint probability distribution relevant to the discrepancy between probability measures. With the inherent uncertainty of the transport plan, we employ entropy language to model a score function, referred to as conditional distribution entropy score function. Then, we discuss the relationship between entropic regularized optimal transport and the proposed conditional distribution entropy score function. Lastly, we give some additional results in Section A.

**Definition 3.4** (Conditional Distribution). *For two discrete random variables $(U, V) \sim \pi(\mu, \nu)$, the conditional distribution of U given value v is defined as,*

$$\pi_{U|V}(u|v) = \frac{\pi_{UV}(u, v)}{\pi_V(v)} \quad \forall u \in dom(U) \tag{4}$$

**Uncertainty Modeling.** In Definition 3.1, it shows that the transport plan $\mathbf{P}$ can be viewed as a joint probability distribution in the form of a matrix. Given a test input, there is a corresponding column in the transport plan, which is essentially a conditional probability distribution, as defined Definition 3.4. Accordingly, the entropy of the conditional probability distribution indicates the level of uncertainty regarding a test input belonging to the OOD. To this end, formally, we define the transport that happens on the test input $v \in dom(V)$ as a random event, represented by a random variable $T$ that follows the conditional distribution, i.e., $T \sim \pi_{U|V}(u|v)$. Then, the score function is denoted as the entropy of conditional distribution $\mathbb{H}(U|V = v)$, as defined in Definition 3.5.

**Definition 3.5** (Conditional Distribution Entropy). *For two discrete random variables $(U, V) \sim \pi(\mu, \nu)$, the entropy of conditional distribution of given value v is defined as,*

$$\mathbb{H}(U|V = v) = - \sum_{u}^{dom(U)} \pi(u|v) log \pi(u|v) \tag{5}$$

With the principle of maximum entropy, the conditional distribution entropy $\mathbb{H}(U|V = v)$ tends to bigger if the corresponding transport plan is more uniform. In other words, it will be a higher chance of being an OOD sample if the given test input has more uncertainty. On the contrary, the entropy value is smaller, if the corresponding transport plan is sparser.

Figure 1 illustrates the uncertainty of transport plan. The training samples are $U = \{u_1\}_{i \leq 6}$ and test inputs are $\{v_j\}_{j \leq 2}$, where $v_1$ is an ID sample and $v_2$ is an OOD sample. It can be observed that the transport from $v_2$ to $\{u_1\}_{i \leq 6}$ is almost uniform. In contrast, the transport

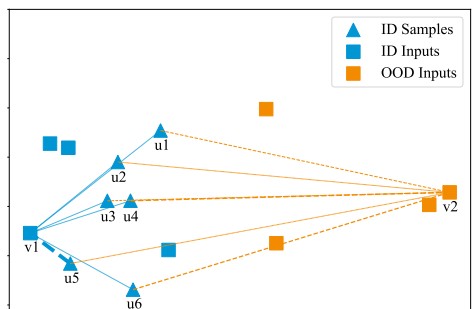

Figure 1: An example of transport plan.

from $v_1$ to $\{u_1\}_{i \leq 6}$ is sparse, since a large portion of the mass is transported from $v_1$ to $u_5$ and a small portion of the mass is transported from $v_1$ to $U - \{u_5\}$. Thus, the uncertainty of the transport from $v_2$ to $U$ is high, whereas the uncertainty of the transport from $v_1$ to $U$ is low.

**Proposition 3.6.** *Let $(U, V) \sim \pi(\mu, \nu)$, when the regularization coefficient $\lambda$ converges to $+\infty$, the conditional entropy of given value $v$, $\mathbb{H}(U|V = v)$ converges to $log|dom(U)|^2$.*

$$\mathbb{H}(U|V = v) \xrightarrow{\lambda \to +\infty} log|dom(U)|$$

Above proposition reveals the inner relation between the entropic regularization OT and the conditional distribution entropy score function. In other words, how the coefficient of entropic regularization $\lambda$ affects the performance of conditional entropic score for OOD detection. As $\lambda$ approaches positive infinity, the performance gradually decreases, where the conditional probability distribution degenerates to a maximum entropy distribution.

### 3.3 CONTRASTIVE TRAINING FOR FEATURE REPRESENTATION

Our method is training-agnostic, it supports both supervised and unsupervised settings. Therefore, we present two kinds feature trainings used in our method, i.e. supervised contrastive training and self-supervised contrastive training (more details see Appendix A). The procedures of our method, including feature extraction and OOD detection are shown in Algorithm 2.

**Supervised Contrastive Loss.** In this work, we employ supervised contrastive training *Sup-Con* (Khosla et al., 2020; Tack et al., 2020) to obtain feature representations. The idea of *SupCon* is to pull together similar samples and push away dissimilar ones.

As shown in Figure 4, the supervised contrastive training consists of three major components, *data augmentation*, *encoder*, and *projection head*. The mechanism of the supervised contrastive training works as follows. For each input sample $(\mathbf{x}_i, y_i), i \in [1, N]$, a pair of augmented samples, i.e., $(\mathbf{x}_i^1, y_i)$ and $(\mathbf{x}_i^2, y_i)$, are generated by the *data augmentation*. Next, the pair of augmented samples are separately fed to the *encoder* $f$, so that a pair of normalized representation vectors $(\mathbf{r}_i^1, \mathbf{r}_i^2)$ are generated. The pair of representation vectors are then mapped into two low-dimensional normalized outputs $(\mathbf{z}_i^1, \mathbf{z}_i^2)$ by the *projection head*. At each iteration, we optimize the following loss function:

$$\mathcal{L}oss = \sum_{i=1}^{2N} -log \frac{1}{|\mathcal{I}(y_i)|} \sum_{k \in \mathcal{I}(y_i)} \frac{e^{\mathbf{z}_i^T \mathbf{z}_k / \tau}}{\sum_{j=1, j \neq i}^{2N} e^{\mathbf{z}_i^T \mathbf{z}_j / \tau}},$$

where $|\mathcal{I}(y_i)|$ is the cardinality of set $\mathcal{I}(y_i)$ representing the indices of all samples except $(x_i, y_i)$. $\tau$ is a scalar temperature parameter.

## 4 EXPERIMENTS

In this section, we evaluate our proposed OOD detection method through extensive empirical studies on multiple ID and OOD datasets. Section 4.1 presents the experimental setup. Section 4.2 reports the empirical studies, which demonstrate that our proposed method achieves state-of-the-art performance on multiple benchmark datasets. Section 4.3 conducts detailed ablation studies and analysis to offer insights into our proposals.

### 4.1 EXPERIMENTAL SETUP

**Datasets.** We evaluate our method on a series of benchmark datasets, including CIFAR-10 (Krizhevsky et al., 2009), CIFAR-100 (Krizhevsky et al., 2009), and SVHN (Netzer et al., 2011). Besides, we conduct experiments on multiple real image datasets, including LSUN (Yu et al., 2015), Place365 (Zhou et al., 2017), Textures (Cimpoi et al., 2014), and tiny ImageNet (Kingma & Welling, 2014).

**Evaluation Metrics.** We use the following metrics to evaluate our method: (1) the false positive rate (FPR95) of OOD samples, when the true positive rate of ID samples is at $95\%$; (2) the area under the receiver operating characteristic curve (AUROC); (3) the area under the precision recall curve (AUPR).

---

[2]Please refer to Appendix A for proof details.

Table 1: Comparison with state-of-the-art methods (including distance-based and non-distance-based methods) on **CIFAR-100 as ID**, and four other datasets as OOD. The best results are in bold. The data of methods are copied from (Ming et al., 2023).

| Method | SVHN | | iSUN | | LSUN | | Textures | | Places365 | |
|---|---|---|---|---|---|---|---|---|---|---|
| | FPR↓ | AUROC↑ | FPR↓ | AUROC↑ | FPR↓ | AUROC↑ | FPR↓ | AUROC↑ | FPR↓ | AUROC↑ |
| | | | | | *Without Contrastive Training* | | | | | |
| MSP(Hendrycks & Gimpel, 2017) | 78.89 | 79.80 | 84.61 | 76.51 | 83.47 | 75.28 | 86.51 | 72.53 | 84.38 | 74.21 |
| ODIN(Liang et al., 2018) | 70.16 | 84.88 | 79.54 | 79.16 | 76.36 | 80.10 | 85.28 | 75.23 | 82.16 | 75.19 |
| Mahalanobis(Lee et al., 2018) | 87.09 | 80.62 | 83.18 | 78.83 | 84.15 | 79.43 | 61.72 | 84.87 | 84.63 | 73.89 |
| Energy(Liu et al., 2020) | 66.91 | 85.25 | 66.52 | 84.49 | 59.77 | 86.69 | 79.01 | 79.96 | 81.41 | 76.37 |
| GODIN(Hsu et al., 2020) | 74.64 | 84.03 | 94.25 | 65.26 | 93.33 | 67.22 | 86.52 | 69.39 | 89.13 | 68.96 |
| SSD(Sehwag et al., 2021) | 70.18 | 80.19 | 83.07 | 68.89 | 81.12 | 73.22 | 59.30 | 82.91 | 89.34 | 64.82 |
| KNN(Sun et al., 2022) | 60.97 | 84.20 | 71.87 | 81.90 | 71.40 | 78.85 | 70.30 | 81.32 | 78.95 | 76.89 |
| **Ours** | **2.42** | **99.49** | **11.69** | **97.36** | **21.88** | **96.12** | **45.35** | **90.76** | **57.24** | **86.38** |
| | | | | | *With Contrastive Training* | | | | | |
| Proxy Anchor(Kim et al., 2020) | 87.21 | 82.45 | 70.01 | 84.96 | 37.19 | 91.68 | 65.64 | 84.99 | 70.10 | 79.84 |
| CE+SimCLR | 24.82 | 94.45 | 66.52 | 83.82 | 56.40 | 89.00 | 63.74 | 82.01 | 86.63 | 71.48 |
| CSI(Tack et al., 2020) | 44.53 | 92.65 | 76.62 | 84.98 | 75.58 | 83.78 | 61.61 | 86.47 | 79.08 | 76.27 |
| CIDER($\alpha = 0.5$)(Ming et al., 2023) | 13.86 | 97.07 | 53.96 | 88.59 | 32.62 | 93.62 | 44.41 | 90.46 | 78.38 | 78.64 |
| SSD+(Sehwag et al., 2021) | 16.66 | 96.96 | 77.05 | 83.88 | 44.65 | 91.98 | 44.21 | 90.98 | 74.48 | 79.47 |
| KNN+(Sun et al., 2022) | 37.26 | 93.12 | 71.58 | 82.48 | 57.97 | 85.63 | 49.60 | 89.10 | 75.53 | 78.44 |
| **Ours** | **12.77** | **97.64** | **30.01** | **94.18** | **9.55** | **98.22** | **38.47** | **92.25** | **52.15** | **89.93** |

**Baselines.** We consider a series of competitors, which can be classified into two categories. The first category consists of models learned without contrastive training, including MSP (Hendrycks & Gimpel, 2017), ODIN (Liang et al., 2018), Energy (Liu et al., 2020), Mahalanobis (Lee et al., 2018), and GODIN (Hsu et al., 2020). The second category consists of the models learned with contrastive training, including Proxy Anchor (Kim et al., 2020), CSI (Tack et al., 2020), CE+SimCLR (Winkens et al., 2020), and CIDER (Ming et al., 2023). Moreover, for a fair comparison, we reproduce two state-of-the-art methods, i.e., SSD and KNN, with ResNet-18 network model following the parameters in (Sehwag et al., 2021; Sun et al., 2022).

**Implementation Details.** We use the same network configurations across all trainings. The simple ResNet-18 (He et al., 2016) architecture was employed as the backbone, which is trained using SGD optimizer with the following settings: a momentum of $0.9$, a weight decay of $10^{-4}$, and a batch size of $512$. The learning rate is initialized as $0.5$ with cosine annealing. The temperature $\tau$ is $0.1$. The dimension of the encoder output is $512$, and the dimensionality of the projection head is $128$. The network is trained for $500$ epochs. We set entropic regularization coefficient $\lambda$ as $0.5$ and $0.1$ for SimCLR and SupCon, respectively.

## 4.2 RESULTS

**Performance Comparison.** We compare the performance of our method with baseline methods, in Table 1. We use CIFAR-100 as ID for training, and the mix of CIFAR-100 testing and a series of other datasets, including SVHN, LSUN, Textures, and Places365, for test inputs. For distance-based method and our method, a light-weighted network backbone, i.e., ResNet18, is used. We can draw two major observations from the results. First, the performance of our proposed method dominates all its competitors. For example, on LSUN, the FPR of our method is merely $21.4\%$ of the second-best method SSD+; on Places365, the AUROC of our method is $10.1\%$ higher than that of Proxy Anchor. Second, the technique of contrastive training helps in improving the performance of OOD detection for a large portion of datasets. For example, on LSUN, the FPR of our method is improved by over $12\%$ by incorporating the contrastive training. Similar observations are drawn on other methods.

Table 2: Unsupervised OOD Detection in AUROC.

| ID OOD | CIFAR-10 (CIFAR-100) | CIFAR-10 (SVHN) | CIFAR-100 (SVHN) |
|---|---|---|---|
| Autoencoder(Hawkins et al., 2002) | 51.3 | 2.5 | 3.0 |
| VAE(Kingma & Welling, 2013) | 52.8 | 2.4 | 2.6 |
| PixelCNN++(Salimans et al., 2017) | 52.4 | 15.8 | - |
| Deep-SVDD(Ruff et al., 2018) | 52.1 | 14.5 | 16.3 |
| Rotation-loss(Gidaris et al., 2018) | 81.2 | 97.9 | 94.4 |
| SSD(Sehwag et al., 2021) | 89.2 | 99.1 | 95.6 |
| **Ours** | **90.0** | **99.2** | **98.4** |

Table 3: Hard OOD Detection Task.

|  | Method | AUROC↑ | AUPR↑ | FPR↓ |
|---|---|---|---|---|
| SimCLR | SSD | 67.25 | 63.30 | 88.72 |
|  | Ours | **79.94** | **74.81** | **79.47** |
| SupCE | KNN | 76.94 | 72.98 | 81.78 |
|  | SSD | 61.89 | 59.34 | 91.00 |
|  | Ours | **80.56** | **75.67** | **79.65** |
| SupCon | KNN+ | 70.10 | 67.83 | 85.91 |
|  | SSD+ | 68.03 | 64.34 | 87.88 |
|  | Ours | **84.22** | **78.95** | **77.00** |

**Unsupervised OOD Detection.** We show the result of unsupervised OOD detection in Table 2. The datasets used for experiments are selected following the setting of (Sehwag et al., 2021). It shows that our method outperforms all its competitors in unsupervised OOD detection. For example, on CIFAR-100 vs. SVHN, the AUROC of our method is $2.8\%$ higher than that of SSD, the second-best method, showing the superiority and generality of our method.

**Performance on Hard Task.** It is known that the OOD detection task is challenging if OOD samples are semantically similar to ID samples (Winkens et al., 2020). We choose the task of CIFAR-100 vs. CIFAR 10, which is commonly adopted as hard task (Ming et al., 2023) for the evaluation of OOD detection algorithms. Then, we evaluate our method on the hard task by comparing with two state-of-the-art distance-based methods, i.e., SSD and KNN. The consistent dominance of our method is observed in all the three training mechanisms, in terms of all evaluation metrics. It shows the superiority of our method in contending with hard task.

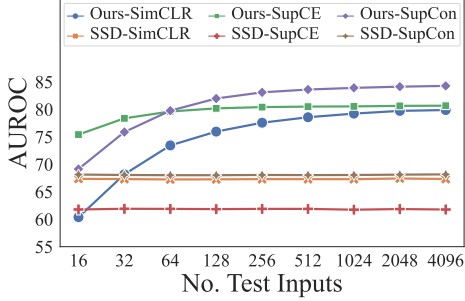

Figure 2: Performance with different number of test inputs on CIFAR-100 vs. CIFAR-10 (in AUROC).

**Performance with different No.Test Inputs.** Out of the consideration of simplicity, we assume the availability of the full test set, which may be a strict setting. Therefore, we divide the test set into a number of equally sized batches and provide the results under different number of test inputs, i.e. No.Test Inputs. As shown in Figure 2, the performance of our method gradually increases and then converges. However, the performance of SSD does not vary with the number of test inputs. It is worth noting that our method outperforms SSD when the number of test samples is over 32 regardless of the training loss, which further demonstrates the superiority of our method.

### 4.3 ABLATION STUDIES

In this section, we study the effect of different components of our proposed method for OOD detection performance. For consistency, all ablation studies are based on the hard task CIFAR-100 vs. CIFAR-10 (in AUROC) under supervised contrastive training.

**Effect of temperature $\tau$.** We study the effect of temperature for the performance of OOD detection. We tune the value of temperature from $0.001$ to $0.5$ for SupCon and SimCLR, as shown in Figure 3 (a). The results show that, with the increase of temperature, the AUROC tends to increase for the supervised contrastive training. However, the temperature has less impact on performance of unsupervised contrastive training.

**Effect of Contrastive Training.** We study the effect of contrastive training on the performance under SupCE and SupCon in Figure 3 (b). The performance is examined by testing on three tasks, CIFAR-10 vs. CIFAR-100, CIFAR-100 vs. CIFAR-10, and CIFAR-10 vs. SVHN. The result shows that SupCon

Figure 3: Ablation studies on CIFAR-100 vs. CIFAR-10 (in AUROC).

outperforms SupCE across all three datasets. To some extent, it shows that the contrastive training helps in improving the OOD detection performance.

**Effect of Training Epoches.** We examine the effect of epochs over the training methods in Figure 3 (c). It shows that, as the increase of epochs, the AUROC of SupCE increases. In contrast, insignificant variation of the AUROC is observed for both SimCLR and SupCon. It implies that the epochs of training do not have a significant effect for methods based on contrastive training.

**Effect of Regularization Coefficient $\lambda$.** In Section 3.2, we have theoretically analyzed how the entropic regularization coefficient $\lambda$ affects the score function. Proposition 3.3 shows that the joint entropy of transport plan would increase until the convergence to a product measure, as the increase of the entropic regularization coefficient $\lambda$. In this case, the entropic score value is a maximal entropy, which is equivalent to a random prediction. Therefore, $\lambda$ should be initialized with a relative small value to obtain a good OOD detection performance. As shown in Figure 3 (d), the trend w.r.t. the increase of $\lambda$ conforms to our theoretical analysis. Also, the observation is consistent for all three training cases, i.e., the performance of OOD detection decreases as the increase of regularization coefficient $\lambda$. In our implementation, we set the value of $\lambda$ as $0.1$ for deployment.

**Effect of Optimal Transport.** We report the effect of optimal transport for OOD detection in Table 4. The performance is examined by using AUROC and FPR as evaluation metrics. As shown in Table 4, optimal transport provides a consistent dominance on the two tasks, i.e., CIFAR-100 vs. CIFAR-10 and CIFAR-10 vs. CIFAR-100, and this fact is observed for both SupCon and

Table 4: Ablation study on optimal transport.

| Dataset | Method | SimCLR | | SupCon | |
|---|---|---|---|---|---|
| | | AUROC↑ | FPR↓ | AUROC↑ | FPR↓ |
| CIFAR-10 vs. | w/o OT | 70.70 | 90.48 | 65.28 | 97.51 |
| CIFAR-100 | with OT | **89.95** | **51.72** | **91.90** | **43.43** |
| CIFAR-100 vs. | w/o OT | 61.96 | 91.55 | 54.35 | 94.55 |
| CIFAR-10 | with OT | **79.94** | **79.47** | **84.22** | **77.00** |

SimCLR. In particular, on the task of CIFAR-10 vs. CIFAR-100, using OT can improve the performance by up to $26.6\%$ on AUROC, and $54.1\%$ on FPR, for SupCon. Even for the hard task of CIFAR-100 vs. CIFAR-10, detecting OOD samples with OT achieves remarkable performance, where the AUROC is steadily above $79\%$. Optimal transport provides a geometric way to differentiate the discrepancy between empirical probability measures. As a result, our approach is essentially utilizing two types of information, i.e., distance information and distributional information. Thus, without optimal transport, our approach may degenerate to the kind of method that uses only distance information, such as KNN.

## 5 CONCLUSION

In this paper, we propose to utilize the discrepancy of empirical distributions for enhancing the performance of OOD detection. We apply discrete optimal transport with entropic regularization to measure the discrepancy between training samples and test inputs. To measure the chance of a test input being an OOD sample, we present a novel conditional distribution entropy score function. We offer theoretical justification and empirical verification to get insights into our proposals. Our method inherits the merit of distance-based method, i.e. parameter-free, training-agnostic, and prior-free. In particular, our method gives prominence in combining the pair- and population-wise information, and therefore offers significant improvement over state-of-the-art OOD detection methods. Extensive experiments on benchmark datasets show the superiority of proposed method.

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

# A  APPENDIX

## A.1  PROOF OF PROPOSITION 3.3

*Proof.* Introducing Lagrangian associated to the primal problem, we have the following dual formulation,

$$\mathcal{L}_\lambda^D(\mathbf{P}, \mathbf{u}, \mathbf{v}) = \langle \mathbf{C}, \mathbf{P} \rangle - \lambda E(\mathbf{P}) - \mathbf{u}^T(\mathbf{P}\mathbf{1}_M - \mu) - \mathbf{v}^T(\mathbf{P}^T\mathbf{1}_N - \nu)$$

Taking partial derivative on transport plan yields,

$$\frac{\partial \mathcal{L}_\lambda^D(\mathbf{P}, \mathbf{u}, \mathbf{v})}{\partial \mathbf{P}_{ij}} = \mathbf{C}_{ij} + \lambda \log \mathbf{P}_{ij} - u_i - v_j = 0$$

$$\mathbf{P}_{ij}^\star = e^{(u_i - \mathbf{C}_{ij} + v_j)/\lambda} = \mathbf{u}_i \mathbf{K}_{ij} \mathbf{v}_j$$

where $\mathbf{u}_i = e^{u_i/\lambda}$, $\mathbf{v}_j = e^{v_j/\lambda}$, and $\mathbf{K} = e^{-\mathbf{C}/\lambda}$. □

## A.2  ADDITIONAL REMARKS

Here, we append the additional remarks for conditional distribution entropy with optimal transport, as mentioned in Section 3.2.

The proposed conditional distribution entropy score function 3.5 is derived from the optimal transport plan, which is the optimal solution of solving entropic regularization OT (Equation 3). As with the regularization item is some joint entropy of transport plan, thus, what is the relation of joint entropy of transport plan and the proposed conditional distribution entropy score function. We show the result in *Remark* A.2.

**Definition A.1** (Conditional Entropy). Given two random variable $(U, V) \sim \pi(\mu, \nu)$, the conditional entropy of $U$ given $V$ is defined as,

$$\mathbb{H}(U|V) = - \sum_u^{dom(U)} \sum_v^{dom(V)} \pi(u, v) \log \pi(u|v)$$

*Remark* A.2 (**Conditional Distribution Entropy and Joint Entropy in OT**). Given two random variable $(U, V) \sim \pi(\mu, \nu)$, following the Definition 3.1 and 3.5, the relation between conditional distribution entropy and joint entropy of transport plan is,

$$\mathbb{H}(U, V) = \sum_v^{dom(V)} \pi(v)\mathbb{H}(U|V = v) + \log dom(V)$$

Above remark shows that we can optimize the joint entropy of transport plan to differentiate OOD samples, which benefits from the entropic regularization OT. The proof is given as follows,

*Proof.* Following the Definition A.1, we rewrite the conditional entropy into the form of conditional distribution entropy,

$$\begin{aligned}
\mathbb{H}(U|V) &= -\sum_u \sum_v \pi(u, v) \log \pi(u|v) \\
&= -\sum_v \pi(v) \sum_u \pi(u|v) \log \pi(u|v) \\
&= \sum_v \pi(v)\mathbb{H}(U|V = v)
\end{aligned}$$

According to the relation between joint entropy and conditional entropy $\mathbb{H}(U, V) = \mathbb{H}(U|V) + \mathbb{H}(V)$, with the uniform weights, we have,

$$\mathbb{H}(U, V) = \sum_v^{dom(V)} \pi(v)\mathbb{H}(U|V = v) + \log dom(V)$$

$\square$

**Conditional Distribution Entropy with OT vs. SSD** SSD (Sehwag et al., 2021) proposes to leverage Mahalanobis distance, calculating the distance between a test input and the centroid of training distribution, as the score function to measure OOD sample, which implicitly incurs multivariate Gaussian distribution assumption(Sun et al., 2022). To relax the distribution assumption and incorporating the more geometric information of feature space, our proposed method is instead to differentiate OOD samples by transporting empirical distribution of test inputs to empirical distribution of training data. In other words, it is to find the optimal mapping at a minimum cost between two empirical distributions (Villani, 2003; Peyré & Cuturi, 2018). Indeed, in a Gaussian context, Mahalanobis distance, formulated as,

$$\mathcal{M}(\mathbf{x}) := \min \ (\mathbf{x} - \boldsymbol{\mu})\boldsymbol{\Sigma}^{-1}(\mathbf{x} - \boldsymbol{\mu})$$

can be viewed some kind of optimal transport that transports data distributed on $\mathcal{N}(\boldsymbol{\mu}, \boldsymbol{\Sigma})$ to a standard spherical normal[3].

### A.3 PROOF OF PROPOSITION 3.6

**Definition A.3** (**Mutual Information**). Given two discrete random variable $U, V$, let $(U, V) \sim \pi(\mu, \nu)$, the mutual information $I(U, V)$ of which is expressed as,

$$I(U, V) = \sum_{u,v} \pi(u, v) log \frac{\pi(u, v)}{\pi(u)\pi(v)}$$

**Lemma A.4.** *The mutual information of two discrete random variables is non-negative, i.e., $I(U, V) \geq 0$.*

*Proof.* To rewrite the formulation of mutual information as

$$I(U, V) = -\sum_{u,v} \pi(u, v) log \frac{\pi(u)\pi(v)}{\pi(u, v)}$$

---

[3]https://math.nyu.edu/ tabak/publications/M107495.pdf

Since the negative logarithm is convex and $\sum_{u,v} \pi(u,v) = 1$, applying the *Jensen Inequality*, we can have,

$$I(U,V) \geq -log \sum_{u,v} \pi(u,v) \frac{\pi(u)\pi(v)}{\pi(u,v)} = 0$$

$\square$

*Proof.* According to the definition of joint entropy 3.1 and the definition of mutual information A.3, the relation between mutual information and joint entropy reads,

$$I(U,V) = \mathbb{H}(U) + \mathbb{H}(V) - \mathbb{H}(U,V)$$

with lemma A.4, we know the joint entropy is bounded as,

$$\mathbb{H}(U,V) \leq \mathbb{H}(U) + \mathbb{H}(V)$$

when $\lambda \to +\infty$, recall the entropic regularization problem $\mathcal{L}_\lambda(\mu, \nu, \mathbf{C})$, it is equivalent to maximize the joint entropy of transport plan, i.e., $\mathbb{H}(U,V)$, and thus,

$$\mathbf{P} \xrightarrow{\lambda \to +\infty} \mu \otimes \nu$$

where the $\otimes$ denotes product measure. In addition, we have uniform weight for marginal measure $\mu$ and $\nu$. Therefore, the entropy of conditional probability distribution in the limit,

$$\mathbb{H}(U|V = v) \xrightarrow{\lambda \to +\infty} log(dom(U))$$

. $\square$

### A.4 PERFORMANCE ON LARGE SEMANTIC SPACE

To evaluate the performance of our method on large semantic space, following the previous works, we choose tiny ImageNet as OOD data. It contains $100k$ data samples in 200 classes. As shown in Table 5, our method outperforms the state-of-the-art distance-based solutions for all three metrics under two training paradigms. In particular, our method improve FPR up to $41\%$ than KNN, the second-best method. In a nutshell, the results show that our method can be adapted well to large-scale OOD detection task.

Table 5: Performance comparison on large semantic space with state-of-the-art methods (CIFAR-100 vs. tiny ImageNet). The best results are in bold.

| Method | AUROC ↑ | AUPR ↑ | FPR ↓ |
|---|---|---|---|
| *without contrastive learning* | | | |
| SSD | 66.06 | 62.26 | 89.87 |
| KNN | 84.04 | 82.43 | 65.54 |
| Ours | **94.71** | **95.01** | **24.42** |
| *with contrastive learning* | | | |
| SSD | 80.18 | 77.39 | 73.14 |
| KNN | 86.26 | 85.56 | 56.86 |
| Ours | **88.34** | **87.79** | **51.16** |

### A.5 COMPUTATIONAL EFFICIENCY

We evaluate the computational efficiency of our method compared to KNN, which also incorporates geometric information of feature space, as shown in Table 6. The reported time overhead is the total time required to detect all test inputs from CIFAR10 ($20k$ images) and SVHN (approximately $36k$ images). When comparing our methods with and without OT to KNN on CIFAR-100 vs. SVHN, we

Table 6: The efficiency comparison with state-of-the-art solution (in $s$).

| Dataset | Method | | |
|---|---|---|---|
| | KNN | Ours with OT | Ours w/o OT |
| CIFAR-100 vs. CIFAR-10 | 66.54 | 64.10 | 37.23 |
| CIFAR-100 vs. SVHN | 114.85 | 106.48 | 49.02 |

achieve $7.2\%$ and $57.3\%$ reduction in computational cost, respectively. On average, processing each test input takes only about 1-2 milliseconds, which is reasonably fast.

**Self-supervised Contrastive Training.** What if training samples are without labels? In practice, it often requires tedious efforts for obtaining high-quality labels. Existing works (Winkens et al., 2020; Hendrycks & Gimpel, 2017) are mostly designed to supervised training[4], it is also important to study unsupervised OOD detection. In the unsupervised setting, the training set $\mathcal{D}_{tr}$ consists of data from the sample space $\mathbb{X}$, without any associated labels, while the test inputs $\mathcal{D}_{te}$ consists of data from $\mathcal{P}_{\mathbb{X}}^{in} \times \mathcal{P}_{\mathbb{X}}^{ood}$. Hereby, we discuss how our proposal can be extended to support the unsupervised setting. Our proposed method roots in the materialized feature space so that the self-supervised training mechanism can be plugged in to train an unsupervised feature extractor. In our implementation, we use *SimCLR* (Chen et al., 2020) for the training, which consists of the same set of components as *SupCon* (Khosla et al., 2020). Due to the absence of labels, it pulls together samples from the same augmentation and vice versa. To do that, we optimize the following loss function,

$$\mathcal{L}oss = \sum_{i=1}^{2N} -log \frac{e^{\mathbf{z}_i^T \mathbf{z}_k / \tau}}{\sum_{j=1, j \neq i}^{2N} e^{\mathbf{z}_i^T \mathbf{z}_j / \tau}},$$

where the numerator compares samples from the same augmentation, yet the denominator compares current sample $\mathbf{z}_i$ with all other samples.

## A.6 THE COMPUTING FRAMEWORK

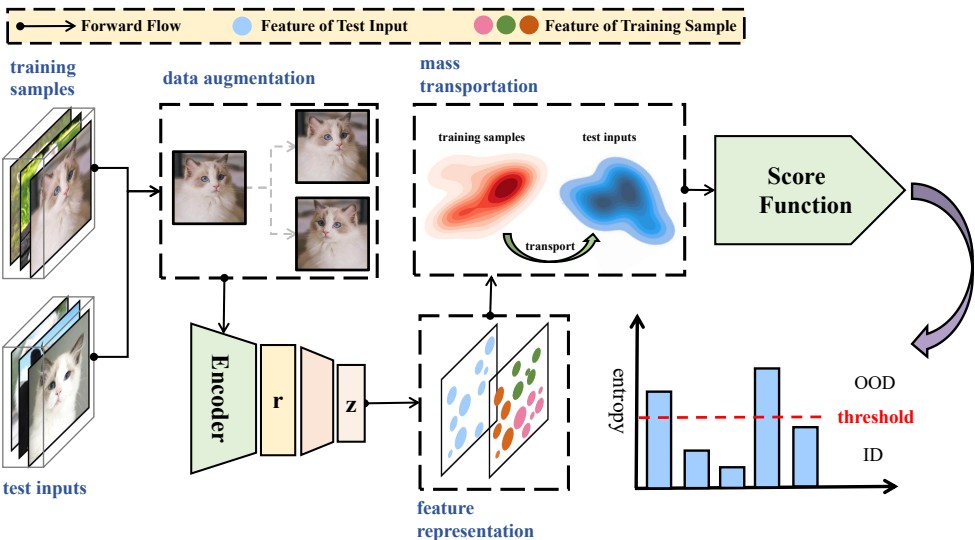

Figure 4: An illustration of framework with OT. The shared encoder receives training samples with data augmentation and test inputs, which produce two kinds of feature representations, forming the corresponding distributions (red area and blue area). After the mass transportation, we can obtain the optimal transport plan, where the conditional distribution entropy of each test input can be derived as score function. A test sample can be identified as ID or OOD by comparing its score with threshold.

---

[4]Some works (Ming et al., 2022; Hendrycks et al., 2019) further require auxiliary OOD data for fulfilling the OOD detection.

## A.7 THE PROCEDURES

---

**Algorithm 2** Detecting OOD Samples via Conditional Transport Entropy with Optimal Transport

---

**Input:** $\mathcal{D}_{tr}, \mathcal{D}_{te}$, feature extractor $f$, projection head $h$, temperature $\tau, \lambda$
**while** training $f$ not converged **do**
  **if** *label is available* **then**
    $\mathcal{L}oss = \sum_{i=1}^{2N} -log\frac{1}{|\mathcal{I}(y_i)|} \sum_{k \in \mathcal{I}(y_i)} \frac{e^{\mathbf{z}_i^T \mathbf{z}_k/\tau}}{\sum_{j=1, j \neq i}^{2N} e^{\mathbf{z}_i^T \mathbf{z}_j/\tau}},$
  **end if**
  $\mathcal{L}oss = \sum_{i=1}^{2N} -log\frac{e^{\mathbf{z}_i^T \mathbf{z}_k/\tau}}{\sum_{j=1, j \neq i}^{2N} e^{\mathbf{z}_i^T \mathbf{z}_j/\tau}},$
**end while**
**for** $i = 1$ to $M$ **do**
  scores $\leftarrow$ computing score using Algorithm 1
**end for**

---

