# OpenReview forum: "Detecting Out-of-Distribution Samples via Conditional Distribution Entropy with Optimal Transport"
_ICLR.cc/2024/Conference — Submitted to ICLR 2024_

### Official Review · Reviewer_MtSB · 2023-11-01

**Soundness:** 2 fair
**Presentation:** 2 fair
**Contribution:** 2 fair
**Rating:** 3
**Confidence:** 4

**Summary:**

The paper proposes a new OOD detection method based on the optimal transport distance between the test input and that of the ID training samples. The transport plan is calculated by standard entropic regularized optimal transport between the empirical distributions of training and test sets. Empirically, the method demonstrates superior OOD detection performance on small-scale benchmarks such as CIFAR-10 and CIFAR-100.

**Strengths:**

- The empirical evaluations and ablations on small-scale benchmarks are comprehensive.

**Weaknesses:**

- The organization and writing of the paper need to be improved. Notations are overly complicated and scattered around. Multiple theorems and definitions can be omitted or miss citations.
  - For example, basic concepts in machine learning such as Definition 3.1 (entropy of two random variables) and Definition 3.4 (condition distribution) can be moved to the Appendix or omitted.
  - Theorem 3.2 should be cited or put as a footnote or remark, as it is a classic result of the entropic regularized optimal transport problem Eq (3). The dual formulation (Proposition 3.3) should also be cited. It might be better to put the standard formulations and results in the Preliminaries Section instead of the Method Section.

   - The formal definition of the OOD detection problem is missing. Therefore, it could be hard to understand why "the OOD detection problem can be formulated as a discrete optimal transport problem with entropic regularization" in P4 (Sec 3.1). Note that the OOD detection problem is a binary classification problem, which is related but **not equal to** solving the optimal transport problem.

  - A transition is needed from Sec 3.2 and Sec 3.3. It is unknown why optimal transport is related to contrastive learning. It seems ill-motivated why "in this work, we employ supervised contrastive training" (Sec 3.3, P6). Is the feature quality of models trained with standard cross entropy loss undesirable?

- The methodology is somewhat vague. It remains unclear how OOD scores are calculated. If I understand correctly, $\nu$ (p3) is the empirical distribution of the validation set, where in practice we only have access to the ID validation set. For OOD inputs, how is the distance calculated?

**Questions:**

- Can authors explain how OOD scores are calculated? As one only has access to the ID validation set, how is the empirical distribution used for OOD detection?

- Why is the method better than KNN or SSD for hard OOD tasks? (Table 3) In principle, it seems that hard OOD samples are challenging for distance-based approaches due to the proximity of OOD samples and ID samples.

---

> ### Author Response · Authors · 2023-11-15
> **Response to Reviewer MtSB**
>
> Thanks for your valuable comments, which helps us improve the manuscript.
>
> ### __Answers to Weaknesses__
>
> __Weakness 1. "The organization and writing of the paper need to be improved. Notations are..."__
>
> __A1.1 & A1.2__ We aim to ensure the paper is self-contained and presented clearly. While there is room for improvement, particularly in the points you highlighted, we plan to enhance these aspects in the revised version.
>
> __A1.3__ Please see the Def 2.1(P2) for the definition of OOD detection.
>
> Our definition aligns with your viewpoint that OOD detection is indeed a binary classification problem. In fact, the entropic regularized OT in our paper is used to model the mass transport between test inputs and ID training data, which is a crucial aspect of our method. Therefore, we agree that it is better to rephrase the sentence that you mentioned to as "we leverage optimal transport with entropic regularization [1] for measuring the discrepancy of empirical distributions between test inputs and ID data." in P4 (Sec 3.1). We will modify it in the revised version.
>
> [1] Cuturi, Marco. "Sinkhorn distances: Lightspeed computation of optimal transport." Advances in neural information processing systems 26 (2013).
>
> __A1.4__ We clarify that our approach is model-agnostic, meaning that the testing procedure is applicable to different model architectures and training losses, including cross entropy loss and contrastive training loss (as shown in Table 1). This has been confirmed in the line of research on distance-based OOD detections. Also, the advantage of contrastive training over cross entropy was discussed in previous works (e.g., KNN and SSD). In our implementation, we employ supervised training, for being in line with existing distance-based OOD detection approaches (e.g., KNN and SSD) and for empirically fair comparison.
>
> We will incorporate the above discussion to make a smooth transition between Sections 3.2 and 3.3.
>
> __Weakness 2. "The methodology is somewhat vague. It remains..."__
>
> __A2.__ Please see Q1 for the calculation of scoring function. The $\nu$ means the empirical distribution of a set of test inputs.
> It is worth noting that our method only uses ID data to train model and obtain training data features without requiring any OOD validation set. Following with distance-based methods (KNN and SSD), we use the trained model to extract the feature of a test input. Then, the distances between a test input and training samples can be calculated by some distance metrics, such as Euclidean distance and cosine distance.
>
>
> ### __Answers to Questions__
>
> __A1.__ We apologize if the calculation of our proposed scoring function has caused confusion.
>
> As detailed in Section 3.2, we introduce a novel scoring function called conditional distribution entropy (Definition 3.5), derived from the optimal transport plan. In this context, we interpret the mass transported from one test input to training data as a conditional distribution, a concept further explained in our discussion on the part of Uncertainty Modeling. Additionally, to quantify the uncertainty of a test input being an OOD sample, we suggest using the entropy of the conditional distribution as the scoring function.
> For an intuitive understanding of the uncertainty in the transport plan, we refer to Figure 1.
> Furthermore, we theoretically explore how entropic regularized optimal transport affects the conditional distribution entropy (Proposition 3.6).
>
> __A2.__ Current challenges faced by existing distance-based methods in handling OOD samples that closely resemble ID in difficult OOD tasks arise from their simplistic utilization of distance information, as seen in methods like SSD or KNN. These approaches rely solely on the distance from test samples to either the centroid of training data or some ID samples.
> In contrast, our approach leverages the empirical distribution of training data and test inputs. By integrating pair-wise distance information with distributional insights through optimal transport, our method develops a scoring function that effectively captures the distinctions between OOD and ID samples. This enables more accurate discrimination of OOD samples that are closely located to ID data.

---

> > ### Comment · Reviewer_MtSB · 2023-11-23
> > **Thanks for the response**
> >
> > Thank you for the responses and clarifications provided. After reviewing the response and considering other reviewers' comments, I find that there are still a few major concerns:
> >
> > > Missing definition of OOD detection
> >
> > While the general definition of OOD detection is given in Def 2.1, the current description seems too general and insufficient to help readers understand the approach.  It would be great to show the concrete formula for OOD detection **based on the optimal transport distance score** beyond the general statement such as "the goal of OOD detection is to identify whether an input x is from ID or OOD".
> >
> > > Vagueness of the method
> >
> > Can authors elaborate on "our method only uses ID data to train model and obtain training data features without requiring any OOD validation set"?  According to Sec 3.1, $\nu$ denotes the discrete empirical probability of the test set $\mathcal{D}_{te}$. At test time, different from prior works that only require access to a single test input, **the whole OOD test datasets are required for the proposed approach to calculate the empirical distribution of the OOD samples**, if I understand correctly. Can authors provide further explanations regarding the requirements for the OOD score calculation?
> >
> > > Writing and organization
> >
> > The organization of the paper currently suffers from redundancy, and the notations used are unclear in places. These issues suggest that a major revision might be necessary to enhance the paper's overall presentation. Also, it appears that the manuscript has not yet been updated to reflect these changes.
> >
> > Given these unresolved issues, it's challenging for me to recommend acceptance of the paper in its current state.

---

> > > ### Author Response · Authors · 2023-11-23
> > > **Thanks for your feedback**
> > >
> > > We plan to update the Uncertainty Modeling as follows, in which the description of OOD detection function can be found.
> > >
> > > __Modified Uncertainty Modeling.__
> > > In Definition 3.1, it shows that the transport plan $\mathbf{P}$ can be viewed as a joint probability distribution $\pi(\mu,\nu)$, with marginals $\mu$ and $\nu$.
> > > Given a transport plan in the form of a matrix, we use two random variables, $U$ and $V$, to describe the randomness of the mass distributed in rows and columns, respectively.
> > > For a test input $v \in dom(V)$, there is a corresponding column in the transport plan, which is essentially a conditional probability distribution $\pi_{U|V}(u|v)$ computed by joint probability $\pi(u,v)$ and marginal probability $\pi(v)$, as defined in Definition 3.4.
> > > Accordingly, the entropy of the conditional probability distribution indicates the level of uncertainty regarding a test input belonging to the OOD. To this end, formally, we define the transport that happens on the test input $v$ as a random event, represented by a random variable $T$ that follows the conditional distribution, i.e., $T\sim \pi_{U|V}(u|v)$.
> > > Then, __the score function is denoted as the entropy of
> > > conditional distribution $\mathbb{H}(U|V=v)$, as defined in Definition 3.5.__
> > >
> > > Following the distance-based methods, our method also trains the model only using ID data and then utilizes the trained model to extract the features of test inputs. Therefore, we don't use OOD data to train the model at all. But we recognize the term "OOD validation set" is inaccurate, as in the response to Reviewer c9Kw.
> > >
> > > Since our proposal is to explore the empirical distribution with geometric information to determine whether it can enhance the OOD detection task, our method requires a set of inputs to build the empirical distribution. As in the response to Reviewer uQxg, we have discussed the limitations and advantages of our proposal in detail.
> > >
> > > We feel sorry about the PDF not yet updated. We will refine the manuscript after the rebuttal period.

---

### Official Review · Reviewer_c9Kw · 2023-11-10

**Soundness:** 2 fair
**Presentation:** 2 fair
**Contribution:** 2 fair
**Rating:** 1
**Confidence:** 5

**Summary:**

The key idea is to employ optimal transport for OOD detection which is especially useful in settings where OOD detection is perform on an entire test set of samples rather than a single test sample. The intuition is that, when estimating Wasserstein distance between a training ID set and a test set, OOD test points will be treated differently from ID test points. While I like the general direction of research, I have problems with details on the approach, especially empirical evaluation and it's similarities to recent works.

Post Rebuttal:
As per my opinion, the authors seem to lack understanding of literature, missing/ignoring many important baselines. Evaluation is far from thorough. The arguments presented in rebuttal are in particular erroneous. For instance, all the well known baselines include (Garg et al, 2023) are unsupervised, not utilizing or assuming knowledge of a validation set when training OOD detector. Using a validation set obtained from ID set itself~(Hendrycks et al., 2019), is a standard practice for tuning the OOD detector so as to maximize OOD detection on OOD validation set while maintaining 5% OOD rate for ID set. While this is an optional step in all the present methods including (Garg et al., 2023), there is no reason to not utilize outlier exposure in any method. Ignoring this ground breaking work by (Hendrycks et al., 2019) as a justification for missing important baselines is misleading to the community. So I am updating my score to strong rejection.

DEEP ANOMALY DETECTION WITH OUTLIER EXPOSURE. Hendrycks et al. ICLR 2019.

**Strengths:**

The general idea of employing information theory for OOD detection, not having to relying upon distributional assumptions, is highly appealing. More such works are required in this direction.

**Weaknesses:**

My concerns are as following. Similar work is accomplished in (Garg et al, 2023) where, instead of Wasserstein distance, KL-divergence is estimated in its dual form between an ID training set and a test set. From that perspective, these two works are highly similar. Especially, in the dual form, expression for Wasserstein distance and optimal transport are similar. While the authors cite the work in the very beginning of the paper in an abstract fashion, the connections between the two works are completely ignored, be it experiments, theory or methodology. Furthermore, as per my understanding, existing methods perform OOD detection for a given test point whereas the proposed method utilizes entire test set. While it is appreciated that, in continual learning settings, a test set is indeed available for better OOD detection rather than restricting, I would have liked to see details on this in the evaluation setup. How do you relate these two different class of methods in same setup? A lot more details are required on this particular aspects. Furthermore, while the framework of optimal transport is appealing despite being a straightforward extension of the previous work (Garg et al, 2023), the other proposed techniques of using contrastive learning, feature extraction, seem unnecessary. As per understanding, as in (Garg et al, 2023), dual function being a neural or kernel similarity like function, would take care of representation learning (feature extraction, contrastive learning), etc. Otherwise, the approach becomes unnecessarily complex, hard to decipher, and looses it appeal to some extent, especially in the era of end to end learning. Last but not least, experimental evaluation should be more thorough in general and it must include experiments on Imagenet as ID dataset as done in the recent works. In particular, the comparison between CIFAR10 vs CIFAR100 as (ID vs OOD) is not intuitive.

**Questions:**

Please see above.

---

> ### Author Response · Authors · 2023-11-15
> **Response to Reviewer c9Kw -- Part I**
>
> Thanks for your insightful comments and approval of the general research direction.
>
> Let's try to summarize your comments into five weak points.
>
> __W1.__ Our method is similar to the method that you have mentioned (Garg et al.).
>
> __W2.__ Our techniques look similar to the techniques of the method that you have mentioned (Garg et al.).
>
> __W3.__ Some evaluation setup details are not clear.
>
> __W4.__ Results on large-scale dataset, e.g., ImageNet.
>
> __W5.__ Reasons of using CIFAR10 vs CIFAR100 as (ID vs OOD).
>
>
>
> ### __Answer to W1__
>
> In fact, our work differs significantly from the study that you have mentioned (Garg et al., 2023) because they operate under different problem settings. The work by Garg et al. (2023) assumes the availability of an OOD validation set. However, it is not practical to assume the explicit knowledge of unknowns due to the vulnerability of OOD inputs [1, 2].
>
> In contrast, our method is distance-based and does not require any OOD validation set. That is why we compared our method with the latest distance-based approaches, such as KNN [3] and SSD [4], as they share the same problem setting but differ from Garg et al. (2023).
>
> We all know it is not fair to compare a method with prior knowledge to one without. But we can compel the comparison to happen, by __using Place365 as OOD validation set for Garg's method.__ The result  shows that our method still outperforms Garg's method in all cases but Place365. __Notice that our method is without the aid of OOD validation set.__
>
> |        | SVHN  |       | ISUN  |       | LSUN  |       | Texture |       | Place365 |       |
> |:------:|:-----:|:-----:|:-----:|:-----:|:-----:|:-----:|:-------:|:-----:|:--------:|:-----:|
> | Method | FPR   | AUROC | FPR   | AUROC | FPR   | AUROC | FPR     | AUROC | FPR      | AUROC |
> | KL     | 83.24 | 70.31 | 42.72 | 91.37 | 75.24 | 80.71 | 88.06   | 70.57 | __45.94__    | __90.24__ |
> | Ours   | __12.77__ | __97.64__ | __30.01__ | __94.18__ | __9.55__  | __98.22__ | __38.47__   | __92.25__ | 52.15    | 89.93 |
>
>
> ### __Answer to W2__
>
> We would like to argue the techniques of our method are significantly different from those in Garg's paper.
>
> __1. The Usage of Statistical Distance.__ The work (Garg et al. 2023) directly uses the dual form of KL divergence, a statistical distance, to estimate the chance of a test input being an OOD sample. In contrast, we don't directly use the Wasserstein distance (another statistical distance) to detect an OOD sample. Instead, we formulate the problem as an entropic regularized OT problem and define a novel scoring function over the optimal transport plan to measure the uncertainty of a test input being an OOD sample.
>
> __2. The Role of Dual Form.__ Despite the involvement of the dual form concept in both papers, it serves a distinct role in each.
> In Garg's work, the dual form of the KL divergence is employed to avoid the estimation of the density function, and the final dual function can be viewed as a function approximator. Differently, our method utilizes the dual problem of entropic regularized OT to efficiently obtain the optimal transport plan. __So, the dual form of our work cannot take care of the representation learning like they did.__
>
>
> ### __Answer to W3__
>
>
> __In Same Setup.__ Our work put an emphasis on utilizing empirical distributions with geometric information for OOD detection, which is overlooked by existing distance-based approaches, such as KNN [3] and SSD [4].
>
> In our experiments, we have tried to evaluate the effect of the cardinality of a test input batch to the performance of OOD detection, as shown in Figure 2. In this setting, an extreme case is to handle a test input as a batch (i.e., batch cardinality is one).
> Here, we provide more results in the table below.
> | No.TestInput (Batch Cardinality)| Method | FPR   | AUROC |
> |:---------:|:------:|:-----:|:-----:|
> | 1         | SSD    | __51.42__ | __88.65__ |
> |           | Ours   | 56.17 | 85.71 |
> | 16        | SSD    | 51.43 | 88.66 |
> |           | Ours   | __40.89__ | __89.58__ |
> | 256       | SSD    | 51.49 | 88.64 |
> |           | Ours   | __30.06__ | __93.53__ |
> | 4096      | SSD    | 51.81 | 88.59 |
> |           | Ours   | __28.80__ | __93.97__ |
>
> The above results show: 1) The empirical distribution information of test inputs is beneficial for OOD detection; 2) As the batch cardinality increases, our method outperforms the distance-based method SSD.
>
> We will incorporate these results in the revised manuscript.

---

> ### Author Response · Authors · 2023-11-15
> **Response to Reviewer c9Kw -- Part II**
>
> __W4.__ Results on large-scale dataset, e.g., ImageNet.
>
> ### __Answer to W4__
> __Evaluation on large-scale ImageNet.__ We take ViT-B/16 as the backbone and follow the parameter settings of KNN. We use ImageNet-1k as the ID dataset and five datasets (in Table 1) as the OOD datasets. The average performance is reported in the table below.
>
> | Model | ID          | Method | FPR   | AUROC |
> |:-----:|:-----------:|:------:|:-----:|:-----:|
> |       |             | SSD    | 22.23 | 94.80 |
> | ViT   | ImageNet-1k | KNN    | 21.39 | 94.70 |
> |       |             | Ours   | __14.34__ | __96.50__ |
>
> The results show the superiority of our proposal over large datasets and models. We will incorporate these results in the revised manuscript.
>
> __W5.__ Reasons of using CIFAR10 vs CIFAR100 (ID vs OOD).
>
> ### __Answer to W5__
>
> In our study, CIFAR10 vs. CIFAR100 are employed in three contexts: hard task, unsupervised detection, and ablation studies. Since classes share multiple similar semantics in these two datasets, they were employed as a challenging task to fully evaluate the OOD detection method in previous works, such as KNN [3], SSD [4], and CIDER [5].
>
> In line with those studies, we chose it for evaluating our method and for comparison with baselines.
>
> #### __References__
>
> [1] Du, Xuefeng, et al. "VOS: Learning What You Don't Know by Virtual Outlier Synthesis." International Conference on Learning Representations. 2021.
>
> [2] Ming, Yifei, Ying Fan, and Yixuan Li. "Poem: Out-of-distribution detection with posterior sampling." International Conference on Machine Learning. PMLR, 2022.
>
> [3] Sun, Yiyou, et al. "Out-of-distribution detection with deep nearest neighbors." International Conference on Machine Learning. PMLR, 2022.
>
> [4] Sehwag, Vikash, Mung Chiang, and Prateek Mittal. "SSD: A Unified Framework for Self-Supervised Outlier Detection." International Conference on Learning Representations. 2020.
>
> [5] Ming, Yifei, et al. "How to Exploit Hyperspherical Embeddings for Out-of-Distribution Detection?." The Eleventh International Conference on Learning Representations. 2022.

---

> ### Author Response · Authors · 2023-11-21
> **Response to Post Rebuttal**
>
> ###  __More Evaluations__
> Given our method's distance-based nature, our comparisons primarily focused on other distance-based methods, including SOTA baselines like KNN and SSD. Notably, the pioneering work by Hendrycks et al. (2019) is not typically regarded as a baseline in current literature discussing distance-based methods, such as KNN and SSD. Consequently, we did not include it in our baseline comparisons.
>
> However, we are open to incorporate more comparisons into our paper if deemed necessary.
> To reproduce OE (Hendery et al. 2019), we use CIFAR100+Tiny ImageNet as ID data, following its settings (See Sec. 4.2.2 & 4.3 in OE).
> We report the performance comparison with OE (Hendery et al. 2019) as follows.
>
> |        | SVHN  |       | iSUN  |       | LSUN  |       | Texture |       | Place365 |       |
> |:------:|:-----:|:-----:|:-----:|:-----:|:-----:|:-----:|:-------:|:-----:|:--------:|:-----:|
> | Method | FPR   | AUROC | FPR   | AUROC | FPR   | AUROC | FPR     | AUROC | FPR      | AUROC |
> | OE     | 71.55 | 84.84 | 68.46 | 82.68 | 78.36 | 82.27 | 79.57   | 76.93 | 79.56    | 78.80 |
> | Ours   | __5.68__  | __98.96__ | __23.28__ | __95.21__ | __5.70__  | __98.90__ | __29.89__   | __94.16__ | __49.26__    | __88.32__ |
>
>
>
>
> ### __Clarity on comments__
> We clarify that the term “OOD validation set” in our comments refer to auxiliary OOD data,  as presented in the seminal work (Hendery et al. 2019). We feel sorry about the confusion and misunderstanding caused by this terminology.
>
> It is known that the auxiliary OOD data mainly includes three categories:
> 1. Generating OOD data from ID sample
> 2. Subset of OOD test data
> 3. Another non-ID and non-OOD-test data
>
> Here, we reclaim the method in that paper (Garg et al. 2023) is significantly different from our method, due to **the usage of OOD data during training**.
>
>
> In the paper (Garg et al. 2023 ), KL dual divergence is computed by training data and the subset of OOD test data, which has been described in P3 as follows: “The key idea is that estimating the divergence measure in its dual form is naturally informative about the subset of samples in the test set that are OOD.”, also affirmed by their [official implementation](https://github.com/morganstanley/MSML/tree/main/papers/OOD_Detection_via_Dual_Divergence_Estimation). In contrast, **our method does not require any auxiliary OOD data during the training stage**.  We guess that this confusion may have led to your misunderstanding.
>
> In particular, the paper by Garg et al. (2023) also utilizes the first type of auxiliary OOD data, i.e., generating OOD data from ID samples, to analyze the generalization of the estimator.
>
> More, your viewpoint: “there is no reason to not utilize outlier exposure in any method.”, is open to discussion in OpenReview.

---

> > ### Comment · Reviewer_c9Kw · 2023-11-22
> > **Please read the literature carefully and then resubmit with proper evaluation**
> >
> > (1) (Hendery et al. 2019) is important one to consider not as a baseline but for their idea of auxiliary OOD data. There is not reason for any model not to use auxiliary OOD dataset, as it requires only the knowledge of ID data to generate it. It is a standard practice in the literature of OOD detection.
> > (2) There are so many baselines that your paper miss or avoided by flimsy arguments on your settings being different from rest of the baselines when it is not.
> > (3) There are standard neural architectures considered in previous works. No reason to completely change your evaluation setup by considering your own choice of architectures.
> > (4) There are 51 OOD test sets considered for Imagenet as the ID set. I don't see any reason to avoid those datasets.
> > (5) Your understanding of the work by (Garg et al. 23) seems erroneous. Please take sometime to read the paper carefully. They do not use OOD data for training. It is an approach based on estimation. Divergence is estimated between an ID training set and a given test set so that test set is split into test ID vs test OOD samples.
> >
> > While I appreciate the back and forth discussion between authors and reviewers, there is value in respecting reviewers' opinions. If not, why even bother to get it reviewed.
> >
> > This is my last comment in regards to this paper. I wish the authors best of luck and sincerely hope that they will put in the required efforts on a thorough experimental validation, and let their brilliant idea mature to be an influential paper in the coming future. To reiterate, I never questioned the proposed idea. I very much appreciate this line of work.

---

> > > ### Author Response · Authors · 2023-11-22
> > > **Thanks for your feedback**
> > >
> > > We appreciate your comments and suggestions. After receiving your comments, we have double-checked the paper and its code.
> > >
> > > First, please allow us to sincerely apologize for our a significant misunderstanding regarding the paper (Garg et al. 2023) since in the main experiment of Garg et al. (2023), only OOD data were employed as the test set (as indicated in Table 1), whereas we utilized a test dataset comprising a mixture of ID and OOD data.
> > >
> > > We recognize the work (Garg et al. 2023) serves as a valuable reference for our study, particularly in terms of method and evaluation. We accept your opinions and agree that two works are similar, as mentioned in your initial comment. Despite there being the same problem settings, it still has differences as we clarified in our previous response.
> > >
> > > Besides, we agree that using auxiliary OOD data is a sort of important idea for detecting OOD samples. However, it seems rather strong to require all methods to use the auxiliary OOD data in their method without any reason. It is worthy noting that not all auxiliary OOD data are obtained from only ID data, as evidenced by the work (Hendery et al. 2019).
> > >
> > > Anyway, we value your insights into the paper (Garg et al. 2023) and willing to refine our paper.
> > > Lastly, we also provide the performance comparisons (the same setting as Table 1 in our paper) with the work (Garg et al. 2023).
> > > | Training Loss | Method | FPR   | AUROC |
> > > |:-------------:|:------:|:-----:|:-----:|
> > > | SupCE         | KL     | 27.82 | 92.41 |
> > > |               | Ours   | __27.71__ | __94.14__ |
> > > | SupCon        | KL     | __14.89__ | 91.11 |
> > > |               | Ours   | 28.58 | __94.04__ |
> > >
> > > The results demonstrate our method is competitive with that work (Garg et al. 2023).

---

### Official Review · Reviewer_uQxg · 2023-11-10

**Soundness:** 4 excellent
**Presentation:** 4 excellent
**Contribution:** 4 excellent
**Rating:** 8
**Confidence:** 4

**Summary:**

This paper proposes to use discrete optimal transport for OOD detection. The arguments are as follows:

In short,
- a test dataset is not OOD if it equal or close in empirical distribution to the training set
- W_p is a measure of distance between two distributions
- That distance is the inf over couplings of a distance between samples , where a coupling is a joint over pairs from two distributions.
- OT algorithms recover this inf coupling which lets one compute a distance
- Under the coupling, given a test point (a point from one distribution) the higher the entropy of the train points (the other distribution), the "less paired" the two distributions are under the couplings
- This means the two distributions are sufficiently different
- The conditional entropy is taken to be a scoring function for OOD detection of an instance from a test set.

**Strengths:**

I really like this paper. While OT has caught on in generative models, it is nice to see it used in distinct but closely related fields like OOD detection.

The presentation of W_p + OT is clean (crisp + no extra details).

The use of the entropy of the transport plan is well motivated.

The numerical experiments are extensive.

**Weaknesses:**

In general the paper is good.

1. (exploring relationship between OT and dim. of encoding in more detail)

Dimension is not discussed too too much in this work, but greatly affects usefulness + ability-to-solve OT and approx. OT such as Sinkhorn. Given that I believe the work is fairly complete as far as other aspects of experiments go (datasets, benchmarks etc) I will choose to push on this. Do you have any experimental results varying the encoder/feature-space dimension? It would be great to know how this affects performance, especially for higher dim image datasets.

2. Could discuss the role of the whole test set in making decisions on each test point, in more detail. See QUESTIONS.



3.  (less a weakness, more a suggestion for helping readers)

If there is a main thing to improve, I think it would be help the not-familiar-with-OT reader gain a little more intuition for higher entropy of the transport plan (resulting from running Sinkhorn, etc) correlating with more-likely-to-be-OOD.

There is a worked example with a few datapoints in figure 1, that is good. You could maybe supplement it and help the reader by pointing out some other visual examples like e.g. if P and Q are same and you have very large N for both, you would get closer to the identity mapping being the solution , which has zero variance/entropy, etc...

In short, I think your typical reader might be quite familiar with OOD but not familiar with OT so you could add just a little more hand-holding in that part of the text since the main intuition for the method comes from understanding why the conditional entropy should be high or low.

**Questions:**

- See Weakness (1) (discussing dimension of encoding)

- Related to the dimension of the encoding is the dimension of the data. Could you clarify all image dataset resolutions? E.g. for LSUN and Imagenet?

-  If I understand correctly, the score for one test point is determined in part by the whole test set's W_p distance to the train set. This is both interesting and raises questions. One analogy is the subset of OOD methods that try to fit a density to the test points to do likelihood ratio tests (the density is evaluated on each test datapoint but is the result of learning on all test points).

If I understand correctly, though you briefly acknowledge using geometry of the whole test set as a motivation, you eventually do not return to a discussion on this particular aspect of the setup. I think it's worth differentiating between methods that use the whole test set for each test point's decision or not. Could you provide more discussion on this (what to take away from it, and what could be inspired by it in the future)? You could potentially consider introducing some new tasks as well such as making decisions on sets of points. Apologies if you discuss this in more detail and I missed it.

- (minor) Could you consider changing the terminology "score function" to "scoring function" or something similar? "Score function" already has a few meanings so for example it's hard to read "conditional distribution score function" and not think of "nabla_u log p(u|v)

---

> ### Author Response · Authors · 2023-11-22
> **Response to Reviewer uQxg**
>
> Thanks for your insightful comments and suggestions, which help us to improve the manuscript.
>
> ### __Answer to Weakness 1__
> Please see answer to Question 1.
>
> ### __Answer to Weakness 2__
> Please see answer to Question 3.
>
> ### __Answer to Weakness 3__
>
> We agree that the readability of paper is very important.
> We will incorporate more visual examples as follows in the revised manuscript (we feel sorry about that we can't provide figure in comments.), to enhance the readability of our paper.
>
> __Showcase.__ When $\mu$ and $\nu$ are identical empirical distributions, the transport of a test input is exactly, suggesting the smallest entropy in the respective conditional distribution.
>
>
>
> ### __Answer to Question 1__
>
> __Dimension of Encoding.__ We provide the ablation study on the dimension of encoding in the table below. Here, we adopt the ResNet18 as the backbone and the same dataset settings as Table 1.
>
> | Backbone    | Feature Dimension | FPR   | AUROC |
> |:--------:|:---------:|:-----:|:-----:|
> | ResNet18 | 128       | 28.25 | 93.79 |
> |          | 256       | 28.31 | 94.50 |
> |          | 512       | 28.58 | 93.94 |
> |          | 1024      | 29.07 | 93.94 |
> |          | 2048      | 33.28 | 92.92 |
> | ResNet50 | 2048      | __22.76__ | __95.11__ |
>
> We can find the effect of feature dimension on performance is not monotonic on a single model. For example, on the backbone ResNet18, the performance decreases when the feature dimension exceeds 1024.
> However, a higher feature dimension yields better performance for larger models, such as ResNet50.
>
>
>
> ### __Answer to Question 2__
> We clarify all test inputs are with the same input dimension of training samples. Thus, we describe the dimension of ID data used in our empirical study.
>
> | ID Dataset | Backbone |Input Dimension |Feature Dimension|
> |:-------------:|:---------:|:---------:|:---------:|
> | CIFAR10/100      |  ResNet18     |     32\*32\*3         |     512        |
> | ImageNet   |   ViT           |     384\*384\*3        |    768          |

---

> ### Author Response · Authors · 2023-11-22
> **Response to Reviewer uQxg**
>
> ### __Answer to Question 3__
> Thanks for your comments.
> As you have pointed out, there are several works (Serrà, Joan, et al. 2019; Schirrmeister, Robin, et al. 2020; Ren, Jie, et al. 2019) that employ the likelihood ratio for OOD detection. Notably, (Serrà, Joan, et al. 2019; Schirrmeister, Robin, et al. 2020;) necessitate training models beyond ID data, requiring additional knowledge, in addition to using multiple samples for making decision on a test point. The work by (Ren, Jie, et al. 2019) doesn't require extra data, but it computes the likelihood ratio by training a background model on perturbed ID data. In contrast, our method doesn't have access to additional data, such as generated data or OOD data, for model training. Despite there being some differences, we agree that it would be an interesting line to consider the likelihood ratio in-depth for OOD detection.
>
> In the sequel, we have tried to discuss the strategy of taking the whole test set in making decisions, covering comparisons with other distance-based competitors, addressing limitations, and contemplating potential future directions.
>
> __Differentiating with methods not using empirical distribution.__ We discussed the relationship between our method and existing distance-based methods. For example, SSD, it uses Mahalanobis distance as a detection metric. As mentioned in page 13, we remark that the Mahalanobis distance can be viewed as a special case of OT. Another baseline KNN, which is a degenerated version of our method (as described in page 9) when OT is eliminated from our method, using the $k$-th distance to ID sample as a detection metric.
>
>
> __Limitations.__ Nevertheless, relying on the entire test input may pose challenges in terms of efficiency, which is the cost paid for performance enhancement. In the current manuscript, we have investigated how the scale of test inputs impacts performance. We split the whole test set into multiple batches with varied batch sizes. As presented in Figure 2, our method benefits from an increase in the size of the test batch and shows a gradual rise in performance until convergence.
>
> Thus, it is of interest to further study the balance between the batch size and the performance variance. It would also give the promise for adapting the techniques to different applications with various performance requirements and affordable computational resources. Also, when test data arrives in batches, another interesting question arises: Can we leverage the information from previously detected batches to enhance OOD detection in subsequent batches?
>
>
> __Future works.__ Shifting our attention to training samples, a mirroring problem arises: how to effectively utilize training samples, which remains underexplored in existing distance-based methods that use all training samples.
>
> One possible solution is to find effective representations in the feature space. Following this line of thought, in our study, one way that we can try to explore these effective representations in the probability space, such as the barycenters of empirical probability distribution.
>
> Another possible solution is to generate virtual samples as effective representations to substitute for all training samples. Certainly, the application of generation is not limited to this. It is known that a kind of research on OOD detection involves accessing known OOD samples during the training stage (OOD exposure). However, assuming the explicit knowledge of unknowns is not practical due to the vulnerability of OOD inputs. This leads to a potential problem: Can we randomly generate virtual OOD samples for OOD exposure?
>
> These discussions can be integrated into a revised manuscript.
>
>
> ####  __References__
>
> Serrà, Joan, et al. "Input Complexity and Out-of-distribution Detection with Likelihood-based Generative Models." International Conference on Learning Representations. 2019.
>
> Schirrmeister, Robin, et al. "Understanding anomaly detection with deep invertible networks through hierarchies of distributions and features." Advances in Neural Information Processing Systems 33 (2020): 21038-21049.
>
> Ren, Jie, et al. "Likelihood ratios for out-of-distribution detection." Advances in neural information processing systems 32 (2019).

---

### Official Review · Reviewer_kQwJ · 2023-11-14

**Soundness:** 3 good
**Presentation:** 3 good
**Contribution:** 3 good
**Rating:** 6
**Confidence:** 3

**Summary:**

This work proposes to perform OOD detection using _conditional distribution entropy_ score, which is the average entropy of the conditional distribution of training data given the test example (Eq 4) under the optimal transport probability (Prop 3.3) which is computed based on the pairwise cosine similarity matrix based on the learned feature representation of a DNN extractor. Furthermore, author proposes to learn the feature representation using supervised contrastive learning (SupCon, Section 3.3). Authors conducted experiments on simple academic benchmarks (CIFAR-100 vs SVHN/iSUN/LSUN/Textures/Places365) using basic architecture (ResNet18), and compared to previous methods with and without contrastive learning, illustrating strong performance. Authors also conducted detailed ablation study on the effect of temperature, contrastive learning, training recipe, and choice of metrics.

**Strengths:**

1. A novel approach to OOD detection via conditional entropy under optimal transport distribution.
2. The technical approach is clearly described with sufficient background information and detailed derivation.
3. The experiment section is done with extensive baselines and academic benchmarks.

**Weaknesses:**

1. The experiments are mostly based on basic architecture and rather stylized academic benchmarks. Experiments on larger models (ResNet-50 / ViT) and more realistic benchmarks (e.g., ImageNet-O) can make the result considerably more convincing.
2. It would be great to see an ablation study of how OOD performance is impacted by the choice of "distance matrix" (C vs P)  and the choice of metrics (probability, entropy, conditional probability, conditional entropy). This may also be related to my question about Table 4 below.
3. In author's description of Section 3.2, there is a slight notation disconnect for how $\pi(u|v)$ / $\pi(u, v)$ / $\pi(v)$ are related to $P^*$ described in previous section. It would be great to spell that out explicitly in terms of how these quantities can be obtained from the matrix (e.g., in the paragraph "Uncertainty Modeling" ).

**Questions:**

Table 4: How are the result from "w/o OT" rows obtained? Specifically, what metric is used for OOD detection?

---

> ### Author Response · Authors · 2023-11-18
> **Response to Reviewer kQwJ -- Part I**
>
> We are encouraged that reviewer finds our work novel, effective, and with extensive experiments.
> Thanks for your valuable comments and suggestions, which we address below:
>
> ### __Answer to weakness 1__
> Thanks for your suggestions. We have added more experimental results for larger models and datasets as suggested.
>
> * __Evaluation on ResNet50.__ Following the experimental settings of SSD, we alter the backbone to ResNet50 and use the CIFAR100 as ID dataset. The average performance over five datasets (in Table 1) is presented in the table below.
>
> |        | SVHN  |       | iSUN  |       | LSUN  |       | Texture |       | Place365 |       |
> |--------|-------|-------|-------|-------|-------|-------|---------|-------|----------|-------|
> | Method | FPR   | AUROC | FPR   | AUROC | FPR   | AUROC | FPR     | AUROC | FPR      | AUROC |
> | SSD+   | 9.63  | 98.14 | 96.48 | 76.89 | 81.00 | 83.49 | 65.79   | 86.71 | 79.04    | 79.03 |
> | KNN+   | 21.49 | 96.00 | 75.67 | 83.07 | 44.53 | 92.32 | 50.37   | 90.13 | 77.57    | 79.40 |
> | Ours   | __5.68__  | __98.96__ | __23.28__ | __95.21__ | __5.70__  | __98.90__ | __29.89__   | __94.16__ | __49.26__    | __88.32__ |
>
>
> * __Evaluation on ImageNet with ViT.__ Following the parameter settings of KNN, we take ViT-B/16 as the backbone. We use ImageNet-1k as the ID dataset and five datasets (in Table 1) as the OOD datasets. The average performance is presented in the table below.
>
> | Model | ID          | Method | FPR   | AUROC |
> |:-----:|:-----------:|:------:|:-----:|:-----:|
> |       |             | SSD    | 22.23 | 94.80 |
> | ViT   | ImageNet-1k | KNN    | 21.39 | 94.70 |
> |       |             | Ours   | __14.34__ | __96.50__ |
>
>
> The above results show the superiority of our method under the settings of larger models and datasets, which is consistent with the result shown in the manuscript. We will incorporate these results in the revised version.
>
>
> ### __Answer to weakness 2__
>
> We agree that doing more ablation studies is good for analyzing and improving the performance, interpretability, and robustness of the work. In the sequel, we provide the results of ablation studies as suggested.
>
> * __Ablation on Distance.__ We enhance our ablation studies by examining the impact of different distance metrics on out-of-distribution (OOD) detection performance. As shown in the table below, it is evident that Euclidean distance and Cosine distance produce comparable results, given their transformability into each other. In general, Euclidean distance served as the cost metric for optimal transport (OT) due, in part, to its well-analyzed theoretical properties. In this paper, we adopt Cosine distance to align with normalized feature training.
>
> | Training Loss | Distance  | FPR   | AUROC |
> |---------------|-----------|-------|-------|
> | SimCLR        | Euclidean | 22.45 | 94.59 |
> |               | Cosine    | 24.86 | 94.43 |
> | SupCon        | Euclidean | 28.40 | 93.48 |
> |               | Cosine    | 28.58 | 94.04 |
>
> * __Ablation on Detection Metric.__  We improve our ablation studies by examining the effect of detection metrics, specifically the scoring function, on out-of-distribution (OOD) detection performance. The 'Max' metric represents the maximum value of the transport plan for each test input. The 'Probability Distance' metric refers to the inner product over the transport plan and distance. The 'Mean' metric indicates the average value of distances between a test input and training samples. The empirical results highlight the efficacy of our uncertainty modeling and underscore the superiority of our proposed conditional distribution entropy scoring function.
>
> | Training Loss | Metric | FPR   | AUROC |
> |---------------|----------------|-------|-------|
> | SimCLR        | Max            | 33.23 | 91.58 |
> |               | Probability Distance       | 29.33 | 93.27 |
> |               | Mean   | 99.28 | 37.97 |
> |               | Conditional Distribution Entropy        | __24.86__ | __94.43__ |
> | SupCon        | Max            | 40.24 | 90.32 |
> |               | Probability Distance       | 34.60 | 92.73 |
> |               | Mean   | 91.60 | 66.17 |
> |               | Conditional Distribution Entropy        | __28.58__ | __94.04__ |
>
> These ablation studies are conducted on the same settings as in Table 1. The reported result is the average performance over five OOD datasets.
> We will incorporate the above results in the revised manuscript.

---

> ### Author Response · Authors · 2023-11-18
> **Response to Reviewer kQwJ -- Part II**
>
> ### __Answer to Weakness 3__
> Thanks. We acknowledge the slight notation discrepancy you pointed out, and we will address it in the revised version as follows:
>
> __Modified Uncertainty Modeling.__
> In Definition 3.1, it shows that the transport plan $\mathbf{P}$ can be viewed as a joint probability distribution $\pi(\mu,\nu)$, with marginals $\mu$ and $\nu$.
> Given a transport plan in the form of a matrix, we use two random variables, $U$ and $V$, to describe the randomness of the mass distributed in rows and columns, respectively.
> For a test input $v \in dom(V)$, there is a corresponding column in the transport plan, which is essentially a conditional probability distribution $\pi_{U|V}(u|v)$ computed by joint probability $\pi(u,v)$ and marginal probability $\pi(v)$, as defined in Definition 3.4.
> Accordingly, the entropy of the conditional probability distribution indicates the level of uncertainty regarding a test input belonging to the OOD. To this end, formally, we define the transport that happens on the test input $v$ as a random event, represented by a random variable $T$ that follows the conditional distribution, i.e., $T\sim \pi_{U|V}(u|v)$.
> Then, the score function is denoted as the entropy of
> conditional distribution $\mathbb{H}(U|V=v)$, as defined in Definition 3.5.
>
>
> ### __Answer to Questions__
> Incorporating optimal transport (OT) into our approach allows us to go beyond traditional distance-based methods that rely solely on the distance information of training samples and test inputs. Our method harnesses OT to capture empirical distribution information in the feature space, enhancing out-of-distribution (OOD) detection.
>
> To assess the impact of optimal transport (OT) on OOD detection performance, we conducted an ablation study. The results are presented in Table 4, where the row labeled (w/o OT) represents calculations based solely on the distance between a test input and training samples. The distance metric employed in this case is the mean Euclidean distance.

---

### Meta-Review · Area_Chair_PCuJ · 2023-12-06

**Metareview:**

In this paper, the authors propose an optimal transport approach to out-of-distribution detection.  They develop a score function they call conditional distribution entropy to measure the uncertainty about whether a test input is out-of-distribution.  This paper had a large variance in reviews with scores of 1, 3, 6 and 8.  The reviewers seemed to agree that the proposed method is technically sound, well-motivated and interesting. The major issues raised by the reviewers were around empirical evaluation, discussion of related work and clarity.

One reviewer (who assigned a score of 1) was concerned in particular with comparison to Garg et al., 2023 and raised concerns regarding the experimental protocols followed.  However, other reviewers didn't seem to share these concerns.  I would point out that paper appeared at UAI, which was shortly before the ICLR deadline.  I think it would be unnecessarily harsh to penalize for not comparing to that work.  Nevertheless, the comparison provided in the author response seems useful to add to the manuscript.

Another concern regarding the empirical evaluation was that the benchmarks and models used were somewhat small and underwhelming.  This would seem like a reasonable concern.  In the author response, the authors did present a significant amount of new experimental results to address this concern.  These experiments and results did seem more compelling (e.g. especially ViT is a large sota model).  However, the reviewers haven't changed their scores and substantial new experiments seems like it would require another round of review.

In my opinion, this paper seems quite close to borderline.  The methodology seems sound and sensible, but the experiments in the paper are a bit too synthetic for what one might consider state-of-the-art research in the OOD subfield.  The new results and proposed changes in the author response seem substantial and could very well put the paper over the bar.  In particular, I find the ViT results to be significantly more compelling and are much closer to the cutting edge of research.  However, incorporating these changes in the final manuscript would be non-trivial, and it seems important to have reviewers vet the new results.  It seems very close, but I would recommend revising the paper and resubmitting to ICML or TMLR.

**Justification For Why Not Higher Score:**

I wouldn't push back against acceptance.  I'm quite on the border for this one.  The paper is sound, but the experiments are ok but underwhelming.  I think it would be much more impactful if the authors incorporated and expanded a bit on the experiments in the author response.  In principle, the authors could do this for the camera ready, but it seems important to have reviewers check such substantial changes.

**Justification For Why Not Lower Score:**

NA

---

### Decision · Program_Chairs · 2024-01-16

Reject